# Neural Program Meta-Induction

**Jacob Devlin**[*]
Google
jacobdevlin@google.com

**Rudy Bunel**[*]
University of Oxford
rudy@robots.ox.ac.uk

**Rishabh Singh**
Microsoft Research
risin@microsoft.com

**Matthew Hausknecht**
Microsoft Research
mahauskn@microsoft.com

**Pushmeet Kohli**[*]
DeepMind
pushmeet@google.com

## Abstract

Most recently proposed methods for *Neural Program Induction* work under the assumption of having a large set of input/output (I/O) examples for learning any underlying input-output mapping. This paper aims to address the problem of data and computation efficiency of program induction by leveraging information from related tasks. Specifically, we propose two approaches for cross-task knowledge transfer to improve program induction in limited-data scenarios. In our first proposal, *portfolio adaptation*, a set of induction models is pretrained on a set of related tasks, and the best model is adapted towards the new task using transfer learning. In our second approach, *meta program induction*, a $k$-shot learning approach is used to make a model generalize to new tasks without additional training. To test the efficacy of our methods, we constructed a new benchmark of programs written in the Karel programming language [17]. Using an extensive experimental evaluation on the Karel benchmark, we demonstrate that our proposals dramatically outperform the baseline induction method that does not use knowledge transfer. We also analyze the relative performance of the two approaches and study conditions in which they perform best. In particular, meta induction outperforms all existing approaches under extreme data sparsity (when a very small number of examples are available), i.e., fewer than ten. As the number of available I/O examples increase (i.e. a thousand or more), portfolio adapted program induction becomes the best approach. For intermediate data sizes, we demonstrate that the combined method of *adapted meta program induction* has the strongest performance.

## 1   Introduction

*Neural program induction* has been a very active area of research in the last few years, but this past work has made highly variable set of assumptions about the amount of training data and types of training signals that are available. One common scenario is example-driven algorithm induction, where the goal is to learn a model which can perform a specific *task* (i.e., an underlying program or algorithm), such as sorting a list of integers[7, 11, 12, 21]. Typically, the goal of these works are to compare a newly proposed network architecture to a baseline model, and the system is trained on *input/output examples* (I/O examples) as a standard supervised learning task. For example, for integer sorting, the I/O examples would consist of pairs of unsorted and sorted integer lists, and the model would be trained to maximize cross-entropy loss of the output sequence. In this way, the induction model is similar to a standard sequence generation task such as machine translation or image captioning. In these works, the authors typically assume that a near-infinite amount of I/O examples corresponding to a particular task are available.

---

[*]Work performed at Microsoft Research.

Other works have made different assumptions about data: Li et al. [14] trains models from scratch using 32 to 256 I/O examples. Lake et al. [13] learns to induce complex concepts from several hundred examples. Devlin et al. [5], Duan et al. [6], and Santoro et al. [19] are able to perform induction using as few one I/O example, but these works assume that a large set of background tasks from the same *task family* are available for training. Neelakantan et al. [16] and Andreas et al. [1] also develop models which can perform induction on new tasks that were not seen at training time, but are conditioned on a natural language representation rather than I/O examples.

These varying assumptions about data are all reasonable in differing scenarios. For example, in a scenario where a reference implementation of the program is available, it is reasonable to expect that an unlimited amount of I/O examples can be generated, but it may be unreasonable to assume that any similar program will also be available. However, we can also consider a scenario like FlashFill [9], where the goal is to learn a regular expression based string transformation program based on user-provided examples, such as ``John Smith → Smith, J.''). Here, it is only reasonable to assume that a handful of I/O examples are available for a particular task, but that many examples are available for other tasks in the same family (e.g., ``Frank Miller → Frank M'').

In this work, we compare several different techniques for neural program induction, with a particular focus on how the relative accuracy of these techniques differs as a function of the available training data. In other words, if technique A is better than technique B when only five I/O examples are available, does this mean A will also be better than B when 50 I/O examples are available? What about 1000? 100,000? How does this performance change if data for many related tasks is available? To answer these questions, we evaluate four general techniques for cross-task knowledge sharing:

- **Plain Program Induction (PLAIN)** - Supervised learning is used to train a model which can perform induction on a single task, i.e., read in an input example for the task and predict the corresponding output. No cross-task knowledge sharing is performed.

- **Portfolio-Adapted Program Induction (PLAIN+ADAPT)** - Simple transfer learning is used to to adapt a model which has been trained on a related task for a new task.

- **Meta Program Induction (META)** - A $k$-shot learning-style model is used to represent an exponential family of tasks, where the training I/O examples corresponding to a task are directly conditioned on as input to the network. This model can generalize to new tasks without any additional training.

- **Adapted Meta Program Induction (META+ADAPT)** - The META model is adapted to a *specific* new task using round-robin hold-one-out training on the task's I/O examples.

We evaluate these techniques on a synthetic domain described in Section 2, using a simple but strong network architecture. All models are fully example-driven, so the underlying program representation is only used to generate I/O examples, and is not used when training or evaluating the model.

## 2 Karel Domain

In order to ground the ideas presented here, we describe our models in relation to a particular synthetic domain called "Karel". Karel is an educational programming language developed at Stanford University in the 1980s[17]. In this language, a virtual agent named Karel the Robot moves around a 2D grid world placing markers and avoiding obstacle. The domain specific language (DSL) for Karel is moderately complex, as it allows if/then/else blocks, for loops, and while loops, but does not allow variable assignments. Compared to the current program induction benchmarks, Karel introduces a new challenge of learning programs with complex control flow, where the state-of-the-art program synthesis techniques involving constraint-solving and enumeration do not scale because of the prohibitively large search space. Karel is also an interesting domain as it is used for example-driven programming in an introductory Stanford programming course.[2] In this course, students are provided with several I/O grids corresponding to some underlying Karel program that they have never seen before, and must write a single program which can be run on all inputs to generate the corresponding outputs. This differs from typical programming assignments, since the program specification is given in the form of I/O examples rather than natural language. An example is given in Figure 1. Note that inducing Karel programs is not a toy reinforcement learning task.

Since the example I/O grids are of varying dimensions, the learning task is not to induce a single trace that only works on grids of a fixed size, but rather to induce a program that can can perform the desired action on "arbitrary-size grids", thereby forcing it to use the loop structure appropriately.

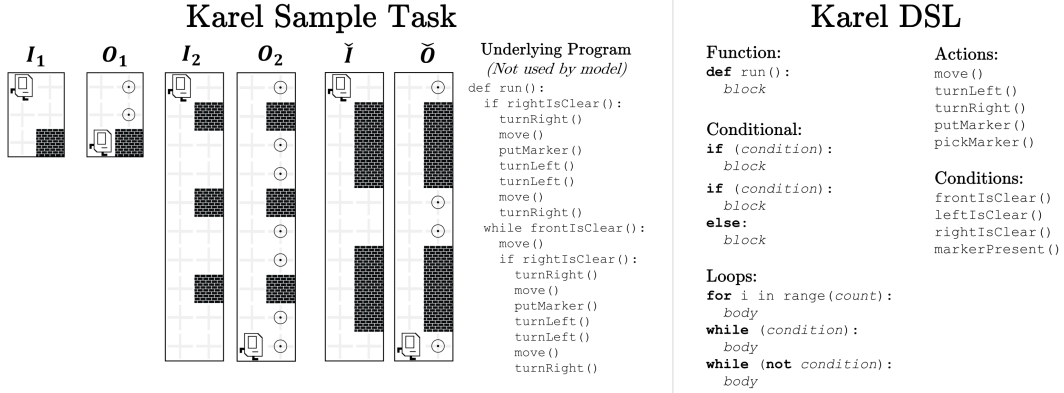

Figure 1: **Karel Domain**: On the left, a sample task from the Karel domain with two training I/O examples $(I_1, O_1), (I_2, O_2)$ and one test I/O example $(\hat{I}, \hat{O})$. The computer is Karel, the circles represent markers and the brick wall represents obstacles. On the right, the language spec for Karel.

In this work, we only explore the induction variant of Karel, where instead of attempting to synthesize the program, we attempt to directly generate the output grid $\hat{O}$ from a corresponding input grid $\hat{I}$. Although the underlying program is used to generate the training data, it is not used by the model in any way, so in principle it does not have to explicitly exist. For example, a more complex real-world analogue would be a system where a user controls a drone to provide examples of a task such as "Fly around the boundary of the forest, and if you see a deer, take a picture of it, then return home." Such a task might be difficult to represent using a program, but could be possible with a sufficiently powerful and well-trained induction model, especially if cross-task knowledge sharing is used.

## 3   Plain Program Induction

In this work, *plain program induction* (denoted as PLAIN) refers to the supervised training of a parametric model using a set of input/output examples $(I_1, O_1), ..., (I_N, O_N)$, such that the model can take some new $\hat{I}$ as input and emit the corresponding $\hat{O}$. In this scenario, all I/O examples in training and test correspond to the same task (i.e., underlying program or algorithm), such as sorting a list of integers. Examples of past work in plain program induction using neural networks include [7, 11, 12, 8, 4, 20, 2].

For the Karel domain, we use a simple architecture shown on the left side of Figure 2. The input feature map are an 16-dimensional vector with $n$-hot encodings to represent the objects of the cell, i.e., (AgentFacingNorth, AgentFacingEast, ..., OneMarker, TwoMarkers, ..., Obstacle). Additionally, instead of predicting the output grid directly, we use an LSTM to predict the *delta* between the input grid and output grid as a series of tokens using. For example, AgentRow=+1 AgentCol=+2 HeroDir=south MarkerRow=0 MarkerCol=0 MarkerCount=+2 would indicate that the hero has moved north 1 row, east 2 rows, is facing south, and also added two markers on its starting position. This sequence can be deterministically applied to the input to create the output grid. Specific details about the model architecture and training are given in Section 8.

## 4   Portfolio-Adapted Program Induction

Most past work in neural programs induction assumes that a very large amount of training data is available to train a particular task, and ignores data sparsity issues entirely. However, in a practical scenario such as the FlashFill domain described in Section 1 or the real-world Karel analogue

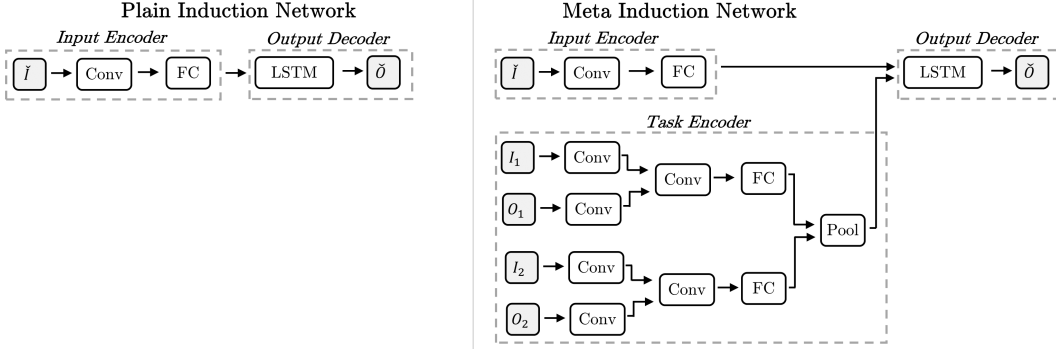

Figure 2: **Network Architecture**: Diagrams for the general network architectures used for the Karel domain. Specifics of the model are provided in Section 8.

described in Section 2, I/O examples for a new task must be provided by the user. In this case, it may be unrealistic to expect more than a handful of I/O examples corresponding to a new task.

Of course, it is typically infeasible to train a deep neural network from scratch with only a handful of training examples. Instead, we consider a scenario where data is available for a number of *background tasks* from the same *task family*. In the Karel domain, the task family is simply any task from the Karel DSL, but in principle the task family can be more a more abstract concept such as "The set of string transformations that a user might perform on columns in a spreadsheet."

One way of taking advantage of such background tasks is with straightforward transfer learning, which we refer to as *portfolio-adapted program induction* (denoted as PLAIN+ADAPT). Here, we have a portfolio of models each trained on a single background I/O task. To train an induction model for a new task, we select the "best" background model and use it as an initialization point for training our new model. This is analogous to the type of transfer learning used in standard classification tasks like image recognition or machine translation [10, 15]. The criteria by which we select this background model is to score the training I/O examples for the new task with each model in the portfolio, and select the one with the highest log-likelihood.

## 5   Meta Program Induction

Although we expect that PLAIN+ADAPT will allow us to learn an induction model with fewer I/O examples than training from scratch, it is still subject to the normal pitfalls of SGD-based training. In particular, it is typically very difficult to train powerful DNNs using very few I/O examples (e.g., $< 100$) without encountering significant overfitting.

An alternative method is to train a single network which represents an entire (exponentially large) family of tasks, and the latent representation of a particular task is represented by *conditioning* on the training I/O examples for that task. We refer to this type of model as *meta induction* (denoted as META) because instead of using SGD to learn a latent representation of a particular task based on I/O examples, we are using SGD to *learn how to learn* a latent task representation based on I/O examples.

More specifically, our meta induction architecture takes as input a set of *demonstration examples* $(I_1, O_1), ..., (I_k, O_k)$ and an additional *eval input* $\hat{I}$, and emits the corresponding output $\hat{O}$. A diagram is shown in Figure 2. The number of demonstration examples $k$ is typically small, e.g., 1 to 5. At training time, we are given a large number of tasks with $k + 1$ examples each. During training, one example is chosen at random to represent the eval example, the others are used to represent the demonstration examples. At test time, we are given $k$ I/O examples which correspond to a *new* task that was not seen at training, along with one or more eval inputs $\hat{I}$. Then, we are able to generate the corresponding $\hat{O}$ for the new task without performing any SGD. The META model could also be described as a $k$-shot learning system, closely related to Duan et al. [6] and Santoro et al. [19].

In a scenario where a moderate number of I/O examples are available at test time, e.g., 10 to 100, performing meta induction is non-trivial. It is not computationally feasible to train a model which is

directly conditioned on $k = 100$ examples, and using a larger value of $k$ at test time than training time creates an undesirable mismatch. So, if the model is trained using $k$ examples but $n$ examples are available at test time ($n > k$), the approach we take is to randomly sample a number of $k$-sized sets and performing ensembling of the softmax log probabilities for each output token. There are ($n$ choose $k$) total subsets available, but we found little improvement in using more than $2 * n/k$. We set $k = 5$ in all experiments, and present results using different values of $n$ in Section 8.

## 6 Adapted Meta Program Induction

The previous approach to use $n > k$ I/O examples at test time seems reasonable, but certainly not optimal. An alternative approach is to combine the best aspects of META and PLAIN+ADAPT, and adapt the meta model to a particular new task using SGD. To do this, we can repeatedly sample $k + 1$ I/O examples from the $n$ total examples provided, and fine tune the META model for the new task in the exact manner that it was trained originally. For decoding, we still perform the same algorithm as the META model, but the weights have been adapted for the particular task being decoded.

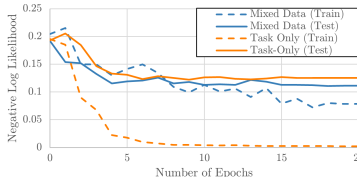

Figure 3: **Data-Mixture Regularization**

In order to mitigate overfitting, we found that it is useful to perform "data-mixture regularization," where the I/O examples for the new task are mixed with random training data corresponding to other tasks. In all experiments here we sample 10% of the I/O examples in a minibatch from the new task and 90% from random training tasks. It is potential that *underfitting* could occur in this scenario, but note that the meta network is already trained to represent an exponential number of tasks, so using a single task for 10% of the data is quite significant. Results with data mixture adaptation are shown in Figure 3, which demonstrates that this acts as a strong regularizer and moderately improves held-out loss.

## 7 Comparison with Existing Work on Neural Program Induction

There has been a large amount of past work in neural program induction, and many of these works have made different assumptions about the conditions of the induction scenario. Here, our goal is to compare the four techniques presented here to each other and to past work across several attributes:

- **Example-Driven Induction** - ✓ = The system is trained using I/O examples as specification. ✗ = The system uses some other specification, such as natural language.
- **No Explicit Program Representation** - ✓ = The system can be trained without any explicit program or program trace. ✗ = The system requires a program or program trace.
- **Task-Specific Learning** - ✓ = The model is trained to maximize performance on a particular task. ✗ = The model is trained for a family of tasks.
- **Cross-Task Knowledge Sharing** - ✓ = The system uses information from multiple tasks when training a model for a new task. ✗ = The system uses information from only a single task for each model.

The comparison is presented in Table 1. The PLAIN technique is closely related to the example-driven induction models such as Neural Turing Machines[7] or Neural RAM[12], which typically have not focused on cross-task knowledge transfer. The META model is closely related are the $k$-shot imitation learning approaches [6, 5, 19], but these papers did not explore task-specific adaptation.

## 8 Experimental Results

In this section we evaluate the four techniques PLAIN, PLAIN+ADAPT, META, META+ADAPT on the Karel domain. The primary goal is to compare performance relative to the number of training I/O examples available for the test task.

| System | Example-Driven Induction | No Explicit Program or Trace | Task-Specific Learning | Cross-Task Knowledge Sharing |
|---|---|---|---|---|
| ***Novel Architectures Applied to Program Induction*** | | | | |
| NTM [7], Stack RNN [11], NRAM [12] Neural Transducers [8], Learn Algo [21] Others [4, 20, 2, 13] | ✓ | ✓ | ✓ | ✗ |
| ***Trace-Augmented Induction*** | | | | |
| NPI [18] | ✓ | ✗ | ✓ | ✓ |
| Recursive NPI [3], NPL [14] | ✓ | ✗ | ✓ | ✗ |
| ***Non Example-Driven Induction (e.g., Natural Language-Driven Induction)*** | | | | |
| Inducing Latent Programs [16] Neural Module Networks [1] | ✗ | ✓ | ✓ | ✓ |
| ***$k$-shot Imitation Learning*** | | | | |
| 1-Shot Imitation Learning [6] RobustFill [5], Meta-Learning [19] | ✓ | ✓ | ✗ | ✓ |
| ***Techniques Explored in This Work*** | | | | |
| Plain Program Induction | ✓ | ✓ | ✓ | ✗ |
| Portfolio-Adapted Program Induction | ✓ | ✓ | ✓ | ✓ *(Weak)* |
| Meta Program Induction | ✓ | ✓ | ✗ | ✓ *(Strong)* |
| Adapted Meta Program Induction | ✓ | ✓ | ✓ | ✓ *(Strong)* |

Table 1: **Comparison with Existing Work**: Comparison of existing work across several attributes.

For the primary experiments reported here, the overall network architecture is sketched in Figure 2, with details as follows: The input encoder is a 3-layer CNN with a FC+`relu` layer on top. The output decoder is a 1-layer LSTM. For the META model, the task encoder uses 1-layer CNN to encode the input and output for a single example, which are concatenated on the feature map dimension and fed through a 6-layer CNN with a FC+`relu` layer on top. Multiple I/O examples were combined with max-pooling on the final vector. All convolutional layers use a $3 \times 3$ kernel with a 64-dimensional feature map. The fully-connected and LSTM are 1024-dimensional. Different model sizes are explored later in this section. The dropout, learning rate, and batch size were optimized with grid search for each value of $n$ using a separate set of validation tasks. Training was performed using SGD + momentum and gradient clipping using an in-house toolkit.

All training, validation, and test programs were generated by treating the Karel DSL as a probabilistic context free grammar and performing top-down expansion with uniform probability at each node. The input grids were generated by creating a grid of a random size and inserting the agent, markers, and obstacles at random. The output grid was generated by executing the program on the input grid, and if the agent ran into an obstacle or did not move, then the example was thrown out and a new input grid was generated. We limit the nesting depth of control flow to be at most 4 (i.e. at most 4 nested if/while blocks can be chosen in a valid program). We sample I/O grids of size $n \times m$, where $n$ and $m$ are integers sampled uniformly from the range 2 to 20. We sample programs of size upto 20 statements. Every program and I/O grid in the training/validation/test set is unique.

Results are presented in Figure 4, evaluated on 25 test tasks with 100 eval examples each.[3] The x-axis represents the number of training/demonstration I/O examples available for the test task, denoted as $n$. The PLAIN system was trained only on these $n$ examples directly. The PLAIN+ADAPT system was also trained on these $n$ examples, but was initialized using a portfolio of $m$ models that had been trained on $d$ examples each. Three different values of $m$ and $d$ are shown in the figure. The META model in this figure was trained on 1,000,000 tasks with 6 I/O examples each, but smaller amounts of META training are shown in Figure 5. A point-by-point analysis is given below:

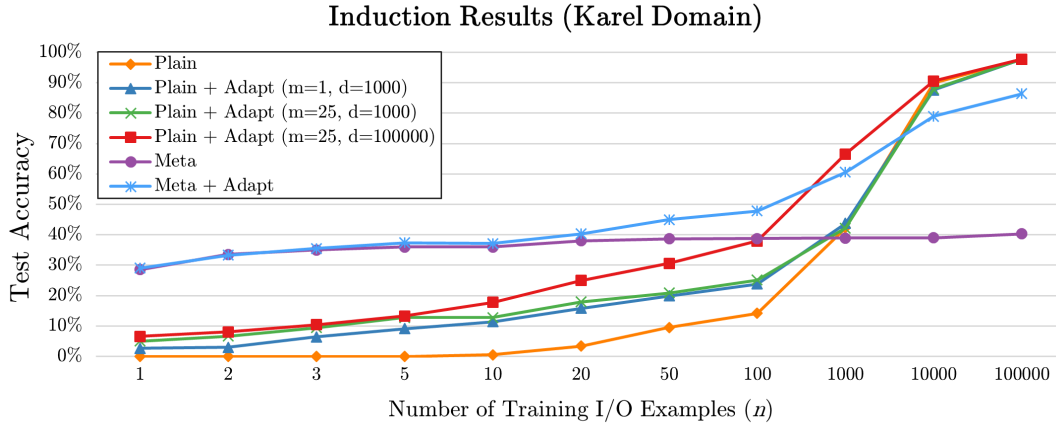

Figure 4: **Induction Results**: Comparison of the four induction techniques on the Karel scenario. The accuracy denotes the total percentage of examples for which the 1-best output grid was exactly equal to the reference.

- **PLAIN** vs. **PLAIN+ADAPT**: PLAIN+ADAPT significantly outperforms PLAIN unless $n$ is very large (10k+), in which case both systems perform equally well. This result makes sense, since we expect that much of the representation learning (e.g., how to encode an I/O grid with a CNN) will be independent of the exact task.

- **PLAIN+ADAPT Model Portfolio Size**: Here, we compare the three model portfolio settings shown for PLAIN+ADAPT. The number of available models ($m = 1$ vs. $m = 25$) only has a small effect on accuracy, and this effect is only present for small values of $n$ (e.g., $n < 100$) when the absolute performance is poor in any case. This implies that the majority of cross-task knowledge sharing is independent of the exact details of a task.

  On the other hand, the number of examples used to train each model in the portfolio ($d = 1000$ vs $d = 100000$) has a much larger effect, especially for moderate values of $n$, e.g., 50 to 100. This makes sense, as we would not expect a significant benefit from adaptation unless (a) $d \gg n$, and (b) $n$ is large enough to train a robust model.

- **META** vs. **META+ADAPT**: META+ADAPT does not improve over META for small values of $n$, which is in-line with the common observation that SGD-based training is difficult using a small number of samples. However, for large values of $n$, the accuracy of META+ADAPT increases significantly while the META model remains flat.

- **PLAIN+ADAPT** vs. **META+ADAPT**: Perhaps the most interesting result in the entire chart is the fact that the accuracy crosses over, and PLAIN+ADAPT outperforms META+ADAPT by a significant margin for large values of $n$ (i.e., 1000+). Intuitively, this makes sense, since the meta induction model was trained to represent an exponential family of tasks moderately well, rather than represent a *single* task with extreme precision.

  Because the network architecture of the META model is a superset of the PLAIN model, these results imply that for a large value of $n$, the model is becoming stuck in a poor local optima.[4] To validate this hypothesis, we performed adaptation on the meta network after randomly re-initializing all of the weights, and found that in this case the performance of META+ADAPT matches that of PLAIN+ADAPT for large values of $n$. This confirms that the pre-trained meta network is actually a *worse* starting point than training from scratch when a large number of training I/O examples are available.

**Learning Curves:** The left side of Figure 4 presents average held-out loss for the various techniques using 50 and 1000 training I/O examples. Epoch 0 on the META+ADAPT corresponds to the META

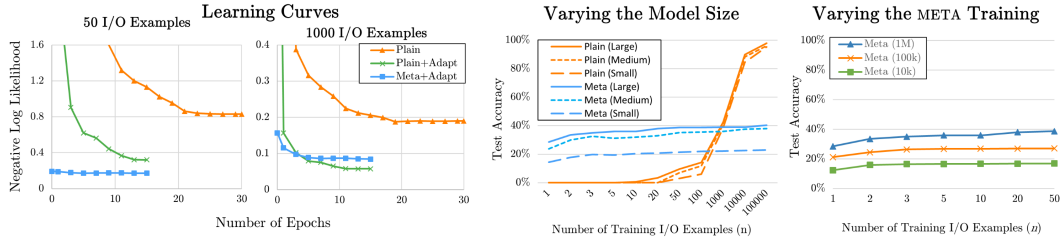

Figure 5: Ablation results for Karel Induction.

loss. We can see that the PLAIN+ADAPT loss starts out very high, but the model able to adapt to the new task quickly. The META+ADAPT loss starts out very strong, but only improves by a small amount with adaptation. For 1000 I/O examples, it is able to overtake the META+ADAPT model by a small amount, supporting what was observed in Figure 4.

**Varying the Model Size:** Here, we present results on three architectures: *Large* = 64-dim feature map, 1024-dim FC/RNN (used in the primary results); *Medium* = 32-dim feature map, 256-dim FC/RNN; *Small* = 16-dim feature map, 64-dim FC/RNN. All models use the structure described earlier in this section. We can see the center of Figure 5 that model size has a much larger impact on the META model than the PLAIN, which is intuitive – representing an entire family tasks from a given domain requires significantly more parameters than a single task. We can also see that the larger models outperform the smaller models for any value of $n$, which is likely because the dropout ratio was selected for each model size and value of $n$ to mitigate overfitting.

**Varying the Amount of META Training:** The META model presented in Figure 4 represents a very optimistic scenario which is trained on 1,000,000 background tasks with 6 I/O examples each. On the right side of Figure 5, we present META results using 100,000 and 10,000 training tasks. We see a significant loss in accuracy, which demonstrates that it is quite challenging to train a META model that can generalize to new tasks.

## 9   Conclusions

In this work, we have contrasted two techniques for using cross-task knowledge sharing to improve neural program induction, which are referred to as *adapted program induction* and *meta program induction*. Both of these techniques can be used to improve accuracy on a new task by using models that were trained on related tasks from the same family. However, adapted induction uses a transfer learning style approach while meta induction uses a $k$-shot learning style approach.

We applied these techniques to a challenging induction domain based on the Karel programming language, and found that each technique, including unadapted induction, performs best under certain conditions. Specifically, the preferred technique depends on the number of I/O examples ($n$) that are available for the new task we want to learn, as well as the amount of background data available. These conclusions can be summarized by the following table:

| Technique | Background Data Required | When to Use |
| --- | --- | --- |
| **PLAIN** | None | $n$ is very large (10,000+) |
| **PLAIN+ADAPT** | Few related tasks (1+) with a large number of I/O examples (1,000+) | $n$ is fairly large (1,000 to 10,000) |
| **META** | Many related tasks (100k+) with a small number of I/O examples (5+) | $n$ is small (1 to 20) |
| **META+ADAPT** | Same as META | $n$ is moderate (20 to 100) |

Although we have only applied these techniques to a single domain, we believe that these conclusions are highly intuitive, and should generalize across domains. In future work, we plan to explore more principled methods for adapted meta adaption, in order to improve upon results in the very limited-example scenario.

## Footnotes

[2]The programs are written manually by students; it is not used to teach program induction or synthesis.

[3]Note that each task and eval example is evaluated independently, so the size of the test set does not affect the accuracy.

[4]Since the DNN is over-parameterized relative to the number of training examples, the system is able to overfit the training examples in all cases. Therefore "poor local optimal" is referring to the model's ability to generalize to the test examples.

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
