[Supplementary Material · nips-2017-meta-supplementary.pdf]

# Supplementary Material for *Neural Program Meta-Induction*

## A   Karel Examples

Figure 1: Four Karel instances from our test set, along with output from the META model. The model generates the correct output for the first three, and the incorrect output for the last instance. The underlying program is shown to demonstrate the complexity of the task, although it it not used by the model.