[Reviews · NeurIPS 2017]

Reviewer 1



The paper is making an evaluation of several approaches for neural network-based induction of computer programs. The main proposal is on the use of meta-learning, in order to exploit knowledge learned from various tasks, for learning on a specific task with a small number of instances. For that purpose, three approaches are proposed: 1) transfer learning for adapting an existing model trained on a related task; 2) meta program induction, where the model has been trained to work on a variety of tasks; and 3) meta program adapted for a given task. The paper also proposes to make use of a synthetic domain Karel, which comes from an educational language for teach computer programming, which consists in moving a robot in a 2D grid through computer instructions. Results are reported with varying the number of instances used for program induction for the three meta approaches proposed + plain method, with results showing some advantages with little number of instances. However, results still are far from good in all cases with little number of instances, with results in the 30%-40% ballpark. The paper is tackling an interesting problem. However, the results and usability of the approaches are not supporting well the proposal. Yes, the meta approaches help the induction process with a very small number of training instances. But still, the results are poor and not clearly of any practical usefulness. Moreover the Karel domain may be convenient for testing program induction, but it appears to be far from practical program induction context. The search space with such setting is very specific, close to the classical maze toy problems of reinforcement learning. Unless for some specific applications (not given in the paper), I am not confident that other program induction setting with behave similarly. The search space of programs for the Karel domain appears to be quite smooth, the program with likely achieve valid results with some random solutions. With most real-life problems, this is not likely to be true, finding valid programs would be by itself challenging. Overall, the paper is well-written, the authors were apparently careful at preparing it. However, the paper stays quite conceptual in its presentation, hindering much low-level details. I found the technical presentation of the proposal (Sec. 8) hard to follow, it tried to pack everything in little space, with explanations not being as precise and detailed as it should. I am still unsure of what the authors did in practice. I am not knowledgeable of the related literature, there are maybe some elements that are obvious for researcher working on that specific problem -- but if this is the case, proper references providing all required background is not properly given. Capacity to reproduce the experiments appears to be very low to me, given that a lot of details are missing. In brief, the proposal in interesting, but the experimental results are not there, so the usefulness of making meta learning for neural program induction has not been demonstrated empirically. Also, the lack of details in the presentation of the technique appears incomplete, while the problem domain tackled is not necessary representative of practical domains for program induction. Therefore, I cannot recommend acceptance of the paper, although I think the core idea of meta-learning for program induction may still be worthful. Stronger experimental evidences supporting the approach are needed to make it acceptable.

Reviewer 2



The paper investigates neural nets learning to imitate programs from varying numbers of input-output examples, both in the plain supervised learning setting and by leveraging transfer across problems. This is done in the domain of Karel, a simple programming language that describes how input grid worlds are transformed into output grid worlds through actions of a so-called hero who moves around in those worlds according to instructions given in a Karel program. The paper randomly generates many such Karel programs and, for each program, a number of corresponding input-output worlds. The learning problem for the neural architectures is to learn from input-output examples how to apply the same transformation to worlds as the (latent) Karel program. The paper compares plain supervised learning, supervised learning that's initialized using the closest previous task, k-shot meta-learning, and k-shot meta-learning that is refined based on the current task. The paper finds that in the Karel domain, each of these techniques has its use depending on the number of tasks and input-output examples per task available. The paper seems technically sound. There is another natural comparison that the authors didn't make, which is to follow the k-shot approach, but to learn a function from the example tasks to the *weights* of a neural net that turns input into output. That is, the task encoder generates the weights of both input encoder and output decoder. In this setting, adaptation is more natural than in the current META+ADAPT case, since you can literally just start with the weights generated by the meta network and optimize from there in exactly the same way that plain supervised learning does. The paper is generally clear, but I don't sufficiently understand the Karel task and the difficulty of the associated learning problem. The paper shows a Karel sample task, but I would have liked to see more examples of the randomly sampled tasks (perhaps in an appendix) so that I can better understand how representative the sample task is. This would have answered the following questions: (1) How complex are the programs sampled from the Karel DSL CFG? A thing that can happen if you sample expressions from recursive PCFGs is that most are really simple, and a few are very complex; the expressions are not of similar size. Is that the case here? (2) How large are the I/O grids? How much variation is there? (3) I'm surprised that the LSTM can output the required diff to transform inputs into outputs as well as it does. If the diffs are long, I would have expected issues with the LSTM "forgetting" the initial context after a while or having trouble to exactly produce the right values or not generalizing correctly. The fact that this works so well makes me wonder whether the learning problem is easier than I think. I was also surprised to see that the NLL is as good as it is (in Figure 5). Because the computational structure of the conv net is quite different from the way the input-output examples are produced by the Karel program (the hero moves around locally to manipulate the grid, the conv net produces the output in one go), I think I would have expected the neural net to only approximately solve the problems. (This impression may well be due to the fact that I don't have a good intuition for what the problems are like.) The table comparing to related work is helpful. That said, I'm still a bit unclear on the extent to which the paper's main contribution is the systematic evaluation of different program induction methods, and to what extent the paper claims to actually have invented some of the methods. I think this paper will be useful to practitioners, conditioned on the results generalizing across domains. As the paper says, the results do seem fairly intuitive, so it seems plausible that they do generalize, but it would certainly have been nice to replicate this in another domain.

Reviewer 3



Neural Program Meta-Induction This paper considers the problem of program induction using neural networks as the model class. Straightforward applications of neural nets require large labelled datasets for the task. The authors propose meta learning to leverage knowledge across different tasks, resembling other meta-learning works such as Santoro et al. The authors further propose that finetuning the model (pretrained on background tasks) on the target task further improves the performance. The paper presents evaluation on a benchmark constructed out of the Karel domain-specific language. The proposed technique closely resembles Santoro’s memory-augmented neural networks[1], and Duan’s one-shot imitation learning[2]. While the latter finds a mention, the former does not. While Duan’s work is specific to imitation learning, Santoro considered the general problem of meta-learning, and evaluate on the omniglot task. It is not clear from the paper how the proposed method is different from MANNs, and whether the distinction is only limited to the choice of benchmark to evaluate on. Santoro and Duan’s models are comfortably applicable to the Karel task, and the proposed model (of this paper) is applicable to omniglot. Thus, the methods could have been compared, and the lack of comparison is conspicuous. It should also be noted that there seems to be no architectural or algorithmic novelty in the proposal which specializes this technique for program induction as opposed to general meta learning/k-shot learning. Program induction often refers to the problem of explicitly learning programs, or at least the model space (even if implicit) being a program (i.e. a sequence of instructions with if-then-else, loops, etc.). The proposed method does not learn a “program”. While the proposed method is a straightforward application of MANNs to a program induction task (which I define as any task where the model is a program), such an application and its empirical evaluation has not been done previously in the literature according to my knowledge (I request the other reviewers to correct me here if I’m wrong). The empirical evaluation on the Karel benchmark compares 4 types of models. PLAIN refers to a model simply trained on input-output pairs for the target task. PLAIN+ADAPT is a model trained on non-target background task (mimicking imagenet-style transfer learning) and then finetuned on the target task. META refers to an LSTM model which takes in training input-output examples for the task, and then predicts output for an unseen task. Thus, the IO examples encode the task, which allows this model to be trained on multiple tasks and thus hopefully take advantage of cross-domain knowledge. META+ADAPT finetunes a META model on the target task. The results show that 1. META+ADAPT is always better than META. 2. PLAIN+ADAPT is always better than PLAIN. 3. META(+ADAPT) is better than PLAIN(+ADAPT) in small dataset regime. 4. PLAIN(+ADAPT) is better than META(+ADAPT) in large dataset regime. The conclusions seem intuitive. To improve the paper further, the authors should make it more clear the contributions of this paper, in particular how it relates to Duan and Santoro's models. A quantitative comparison with those models in common domains of applicability (such as omniglot) would also be helpful. Finally, program induction-specific architectures and algorithms would help in the novelty factor of the paper. [1]: https://arxiv.org/abs/1605.06065 [2]: https://arxiv.org/abs/1703.07326 -- Post Rebuttal -- Thanks for your response. It would be very valuable of Karel task was open-sourced so that the community could make further progress on this interesting task. Please also add citations to Santoro et al. and discuss the relationship of your model with Santoro et al. I increase the score by 1 point.